



## Streamflow estimation at partially gaged sites using multiple

## dependence conditions via vine copulas

Kuk-Hyun Ahn[1]

[1]Assistant Professor, Department of Civil and Environmental Engineering, Kongju National
University, Cheon-an, South Korea; *Corresponding author;* e-mail: ahnkukhyun@gmail.com



26                                    ABSTRACT

Reliable estimates of missing streamflow values are relevant for water resources planning and
management. This study proposes a multiple dependence condition model via vine copulas for
the purpose of estimating streamflow at partially gaged sites. The proposed model is attractive
in modeling the high dimensional joint distribution by building a hierarchy of conditional
bivariate copulas when provided a complex streamflow gage network. The usefulness of the
proposed model is firstly highlighted using a synthetic streamflow scenario. In this analysis,
the bivariate copula model and a variant of the vine copulas are also employed to show the
ability of the multiple dependence structure adopted in the proposed model. Furthermore, the
evaluations are extended to a case study of 54 gages located within the Yadkin-Pee Dee River
Basin, the eastern U. S. Both results inform that the proposed model is better suited for infilling
missing values. After that, the performance of the vine copula is compared with six other
infilling approaches to confirm its applicability. Results demonstrate that the proposed model
produces more reliable streamflow estimates than the other approaches. In particular, when
applied to partially gaged sites with sufficient available data, the proposed model clearly
outperforms the other models. Even though the model is illustrated by a specific case, it can be
extended to other regions with diverse hydro-climatological variables for the objective of
infilling.
**Keywords: vine copulas, multiple dependence condition model, streamflow estimation**

**and infilling approach**




## 1. Introduction

Hydrological observation records covering long-term periods are instrumental in water resources planning and management including the design of flood defense systems and irrigation water management (Aissia et al., 2017; Beguería et al., 2019). However, available streamflow data is often limited due to several situations like equipment failures, budgetary cuts, and natural hazards (Kalteh and Hjorth, 2009). Missing data is particularly observed in remote catchments where equipment failures are repaired only after significant delays following extreme events, which can be crucial for hydrological frequency analysis. Hence, hydrologists often rely on simulated sequences to infill missing data in partially gaged catchments (Booker and Snelder, 2012) by using two primary modeling approaches such as: (1) process-based models (i.e., estimating streamflow based on a conceptual understanding of hydrological processes), and (2) transfer-based statistical models (i.e., transferring information from gaged to ungagged catchments) (Farmer and Vogel, 2016). This paper focuses on the latter, which estimates historical daily streamflow at inadequately and partially gaged sites by the means of a statistical relationship.

Over the past few decades, a variety of statistical models including simple drainage area scaling (Croley and Hartmann, 1986), spatial interpolation technique (Pugliese et al., 2014), regression model (Beauchamp et al., 1989) and flow duration curves (FDCs; Hughes and Smakhtin, 1996), have been developed. In particular, the flow duration curve method has been regarded as one of the most trustworthy regionalization approaches (Archfield and Vogel, 2010; Boscarello et al., 2016; Castellarin et al., 2004; Li et al., 2010; Mendicino and Senatore, 2013). If the target watershed is completely ungaged, FDCs can be established using regression models to





regionalize the parameter sets of defined distributions (e.g., Ahn and Palmer, 2016a; Blum et
al., 2017) or to regionalize a set of primary quantiles (Cunderlik and Ouarda, 2006; Schnier
and Cai, 2014; Zaman et al., 2012). On the other hand, if the target watershed is poorly or
partially gaged, FDC models are built using the following four steps: (1) estimating non-
exceedance probability for recorded streamflow from the target watershed of interest; (2)
selecting one or multiple donor watersheds for the target watershed; (3) transferring the time-
series of non-exceedance probability from the donor watershed(s) for missing streamflow
values; and (4) converting corresponding streamflow values back from the transferred non-
exceedance probability. When FDCs are utilized for partially gaged watersheds, how the donor
watersheds are selected (step 2) and how the probabilities are transferred from the donor
watersheds (step 3) play crucial roles in the FDC framework.

Many studies have developed diverse approaches for steps 2 and 3 in FDC modelling. While
the basic formulation is that non-exceedance probabilities of the target site are transferred by
those at the single donor site, a weighted average of non-exceedance probability from the
selected donor sites has been suggested by Smakhtin (1999) instead. In addition, Farmer (2015)
adopted a kriging model to regionalized daily standard (i.e., z-scored) probabilities based on
non-exceedance probabilities from many donors in a region, using the quantile function of a
standard normal distribution. Although these studies are promising, the joint distribution of
non-exceedance probability between the target and donor watersheds is modeled based on a
Gaussian assumption which cannot properly permit different percentile values such as extremes
that have different spatial dependence structures from donor sites. To circumvent this limitation,
Worland et al. (2019) suggested the copula theory after showing that a unifying framework of
copulas is equivalent to that of FDC (i.e., estimations of the conditional probabilities at the





target watershed given known values at the donors).

Increasing attention has been paid to copulas in the field of hydrology, with applications in
flood frequency analysis, drought risk analysis, and multi-site streamflow simulations (Ahn
and Palmer, 2016b; Ariff et al., 2012; Chen et al., 2015; Daneshkhah et al., 2016; Fu and Butler,
2014). Copulas are efficient mathematical functions that are capable of combining univariate
marginal distribution functions of random variables into their joint cumulative distribution
function and allow representation of diverse dependence structures between these random
variables corresponding to their family members (Sklar, A., 1959). For example, Fu and Butler
(2014) showed that the Gumbel copula performs well in representing multiple flooding
characteristics as compared to the other copulas from the Archimedean family, namely the
Clayton and Frank copulas. To estimate streamflow (i.e., infilling missing data) at poorly and
partially gaged sites, Worland et al. (2019) have developed bivariate copulas with an
Archimedean copula, but limited their application to a single donor. Albeit the limitation, their
bivariate copulas may be acceptable since the higher dimension of copulas is not rich enough
to model all possible mutual dependencies among multisite donors (see Karmakar and
Simonovic, 2009 for details). Hao and Singh (2013) also describe that multivariate copulas are
incapable of modeling multisite data exhibiting complex patterns of dependence.

However, if the theoretical limitation of a multivariate copula is mitigated, dependency
information from multiple donor sites may allow more reliable predictions of regionalized
streamflow. Vine copulas, also known as pair copulas, offer a far efficient way to construct
higher dimensional dependence (Bedford et al., 2002; Joe, 1997). They have hierarchical



structures that sequentially apply bivariate copulas as the building local blocks for constructing
a higher dimensional copula. The high flexibility of vine copulas enables modeling a wide
range of complex data dependencies. In particular, Aas et al. (2009) have popularized two
classes of vine copulas, canonical vines (C-vines) and drawable vines (D-vines) by allowing
diverse pair-copula families such as the bivariate Student-t copula and bivariate Clayton copula.
After the seminal paper, those two vines have been used in many fields including economics
(Arreola Hernandez et al., 2017; Zimmer, 2015), finance (Dissmann et al., 2013; Lu, 2013),
and engineering (Bhatti and Do, 2019; Erhardt et al., 2015; Xu et al., 2017). Similarly, a few
studies have used vine copulas in hydrologic applications with diverse purposes (Daneshkhah
et al., 2016; Liu et al., 2015; Vernieuwe et al., 2015) although they have not been introduced to
infill missing data.

Based on the usefulness of vine copulas, Kraus and Czado (2017) have developed a promising
algorithm that sequentially fits such a D-vine copula model ($\mathcal{M}_{\mathrm{Kraus}}$). The algorithm adds
covariates to the model with the objective of maximizing a conditional likelihood and stops
adding covariates to the model when none of the remaining covariates can significantly
increase the model's conditional likelihood. While it is promising, one challenge that can arise
but has not been previously discussed is overfitting when covariates are correlated with each
other. In this situation, the model may adopt ineffective covariates and eventually leads to poor
predictions. In particular, for the purpose of infilling, streamflow values at the target site are
often correlated by those of many donors. Although the structure of $\mathcal{M}_{\mathrm{Kraus}}$ is potentially
favorable to estimate streamflow, modified model procedure is required to determine the most
influential covariates.




This study forwards two novel contributions to infill missing data in the field of hydrology: (1)
a D-vine copula-based model is introduced to estimate streamflow for poorly and partially
gaged watersheds and (2) the existing model ($\mathcal{M}_{\text{Kraus}}$) is further improved by incorporating a
new procedure to determine the optimal number of donor sites (namely $\mathcal{M}_{\text{Dvine}}$). First,
synthetic data are generated to compare $\mathcal{M}_{\text{Kraus}}$ and $\mathcal{M}_{\text{Dvine}}$. In this analysis, bivariate
copulas (namely $\mathcal{M}_{\text{Bicop}}$) is also employed to demonstrate the usefulness of a high
dimensional joint dependence structure. Afterwards, a real infilling example is utilized to
compare the proposed vine-based model with six other streamflow-transfer models adopted in
literatures.

**2. Methodology**
*2.1 D-vine copulas*
A copula $C$ is $k$-variate cumulative distribution function on $[0,1]^k$ with all uniform margins.
The $C$ can be understood as a function that links the marginal cumulative distributions
$(F_1, \dots, F_k)$ to form a joint distribution $F$. The $C$ associated with joint distribution $F$ is a
distribution function $C: [0,1]^k \rightarrow [0,1]$ such that, for all streamflow vector $\boldsymbol{q} =$
$(q_1, \dots, q_k)^T$, the $C$ satisfies:

$$F(q_1, \dots, q_k) = C(F_1(q_1), \dots, F_k(q_k)) \qquad\qquad \text{Eq. (1)}$$



where $C$ is unique if $F_1, \dots, F_k$ are continuous.
Based on Sklar's theorem (Sklar, A., 1959), a multivariate distribution function is a
composition of a set of marginal distributions; thus, equation (1) can be expressed in terms of
densities,

$$f(q_1, \dots, q_k) = [\textstyle\prod_{i=1}^{k} f_i(q_i)] c(F_1(q_1), \dots, F_k(q_k)) \qquad \text{Eq. (2)}$$


where $c$ is a $k$-dimensional copula density acquired by partial differentiation of the copula $C$
(i.e., $c\big(F_1(q_1), \dots, F_k(q_k)\big) := \frac{\partial^k}{\partial_1 \cdots \partial_k} C\big(F_1(q_1), \dots, F_k(q_k)\big)$) and $f_i(\cdot)$ is the marginal density
corresponding to $F_i(\cdot)$.

Following Bedford and Cooke (2001), any copula density $c(F_1(q_1), \dots, F_k(q_k))$ can be
decomposed into a product of $k(k-1)/2$ pair copula densities. Aas et al. (2009) adopted this
idea and introduced the copula class of pair copula constructions (PCCs) known as vine copulas.
These copulas are suitable to model various dependency structures. Vine structures established
by $k(k-1)/2$ pair copulas are arranged in $k-1$ trees (Brechmann et al., 2013) and can be
categorized as C-vines and D-vines (Liu et al., 2015). This study focuses on D-vines since they
are more widely used in practice (Daneshkhah et al., 2016).

A D-vine is characterized by the ordering of its variables (see Figure 1). In the first tree, the





dependence of the first and second variables, of the second and third, of the third and fourth,
and so on, is modeled using pair-copulas. In the second tree, conditional dependence of the first
and third given the second variable (i.e., $c_{1,3|2}(F(q_1|q_2), F(q_3|q_2))$), the second and fourth
given the third (i.e., $c_{2,4|3}(F(q_2|q_3), F(q_4|q_3))$), and so on, is modeled. Similarly, pairwise
dependencies of two variables are modeled in subsequent trees conditioned on those variables
which lie between the two variables in the first tree (e.g.,
$c_{1,5|2,3,4}(F(q_1|q_2,q_3,q_4), F(q_5|q_2,q_3,q_4))$). The density of the $k$-dimensional D-vine can be
computed as follows (Aas et al., 2009):

$$f(q_1, \dots, q_k) = [\textstyle\prod_{i=1}^{k} f_i(q_i)] \times$$
$$\textstyle\prod_{j=1}^{k-1} \prod_{\acute{j}=1}^{k-j} c_{j,j+\acute{j}|(\acute{j}+1):(j+\acute{j}-1)}(F(q_{\acute{j}}|q_{\acute{j}+1}, \dots, q_{\acute{j}+j-1,}), F(q_{\acute{j}+j}|q_{\acute{j}+1}, \dots, q_{\acute{j}+j-1,})) \quad \text{Eq. (3)}$$

where $c_{j,j+\acute{j}|(\acute{j}+1):(j+\acute{j}-1)}$ indicates the bivariate copula densities.
For the five-dimensional D-vine copula as an example in Figure 1, the corresponding vine
distribution has the joint density as follows:

$$f(q_1, \dots, q_5) = [\textstyle\prod_{i=1}^{5} f_i(q_i)] c_{12} \cdot c_{23} \cdot c_{34} \cdot c_{45} \cdot c_{13|2} \cdot c_{24|3} \cdot c_{24|3} \cdot c_{35|4} \cdot c_{14|23} \cdot c_{25|34} \cdot$$
$$c_{15|234} \hspace{8cm} \text{Eq. (4)}$$

where $c_{1,2}(F_1(q_1), F_2(q_2))$ is simply denoted as $c_{1,2}$.




As presented in equation (4), the conditional distribution functions and conditional bivariate

copulas are required in vine copula modeling. The conditional distribution functions

$F(q_{\dot{j}}|q_{\dot{j}+1}, \dots, q_{\dot{j}+j-1})$, also known as $h$-functions, in equation (4) can be addressed using the

pair-copulas from lower trees by using equation (5). Let $q_i$ be a conditional value of

$q_{\dot{j}+1}, \dots, q_{\dot{j}+j-1}$ and $\boldsymbol{v} = \{q_{\dot{j}+1}, \dots, q_{\dot{j}+j-1}\}\backslash q_i$ the streamflow vector without $q_i$ used in the

following recursive relationship (Aas et al., 2009):

$$h\big(q_{\dot{j}}\big|\boldsymbol{v}\big) := F\big(q_{\dot{j}}\big|\boldsymbol{v}\big) = \frac{\partial C_{\dot{j}i|\boldsymbol{v}}(F(q_{\dot{j}}|\boldsymbol{v}),F(q_i|\boldsymbol{v}))}{\partial F(q_i|\boldsymbol{v})} \qquad \text{Eq. (5)}$$

where the $h$-function is associated with the pair-copula $C_{\dot{j}i|\boldsymbol{v}}$.

More details about D-vines can be found in Bedford et al., (2002) and Czado (2010, 2019).

*2.2 Algorithm of D-vine copula-based estimation ($\mathcal{M}_{\text{Dvine}}$)*

Following Kraus and Czado (2017), a two-step estimation procedure is adopted for the

prediction of the streamflow value at the target watershed. The algorithm ($\mathcal{M}_{\text{Dvine}}$) is

developed using two library packages in the R programming language (Bevacqua, 2017;

Schepsmeier et al., 2015).






Let $q_k$ be the quantile of streamflow at the target watershed given the streamflow values
$q_1, \dots, q_{k-1}$ from the donor sites. In the first step, the marginal cumulative probabilities
$F_k(q_k)$ and $F_j(q_j)$, $j = 1, \dots, k-1$, are estimated using the semiparametric approach. To be
specific, this study uses the continuous kernel smoothing estimator (Geenens, 2014), which is,
given observed streamflow $q_i^\zeta$, $\zeta = 1, \dots, \xi$, at $i$th site, defined as $\widehat{F}_i(q_i) = \frac{1}{nh} \sum_{\zeta=1}^{\xi} \Omega(\frac{q_i - q_i^\zeta}{h})$.
Here, $\Omega(q_i)$ is the "kernel" function with $\omega(\cdot)$ being a symmetric probability density
function and $h$ is the parameter controlling the smoothness of the final estimate. In this study,
a Gaussian kernel is used for all $\omega(\cdot)$. The estimated cumulative probabilities are then
employed to model the D-vine copula in the second step.

Next, to easily estimate conditional streamflow values at the target site, the D-vine copula is
fitted with fixed order $F_k(q_k) - F_{I_1}(q_{I_1}) - F_{I_2}(q_{I_2}) - \dots - F_{I_{k-1}}(q_{I_{k-1}})$, such that $F_k(q_k)$ is
the first node in the first tree and the other orders of donors ($I_1$, …, $I_{k-1}$) are decided based
on their correlations to the target site (i.e., $F_{I_1}(q_{I_1})$ showing the greatest correlations to
$F_k(q_k)$). To build the D-vine copula model, five bivariate copulas (Gaussian, Student-t, Frank,
Gumbel, and Clayton copulas) are considered as potential pair copulas (building blocks) to
represent diverse dependence structures. For example, a Gaussian copula is proper when the
non-exceedance probabilities between two watersheds are associated in the body of their
distribution but are not asymptotically dependent in the both tails. On the other hand, a Gumbel
copula may be appropriate for the situation wherein the non-exceedance probabilities exhibit
tail dependence, where high flows are connected by same rainfall events but low flows are not
correlated (e.g., due to regulation) (Salvadori and De Michele, 2004). Details of the five
bivariate copulas are presented in the Supporting Information. Parameters for the five bivariate



copulas are estimated based on Kendall rank-based correlation ($\rho^\tau$) between sites. The optimal
bivariate copula for each pair copula is determined based on the panelized likelihood function
(i.e., AIC).

The final number ($\chi_k$) of donor sites is further optimized under a cross-validation approach. In
this approach, 80 % of the regional data are employed for model fitting; the other 20 %, for
testing. Again, this procedure is conducted 5 times, each time using a different set of data for
testing. As a measure for the model's fit, the root mean squared error (RMSE; equation (6))
from observed streamflow at the target site is utilized.

$$RMSE_{\chi_k} = \sqrt{\frac{1}{\xi}\sum_{\zeta=1}^{\xi}(q_k - \hat{q}_k^\chi)^2}$$    Eq. (6)

Finally, conditional streamflow values at the target site can be estimated using the inverse form
of the conditional distribution function (i.e., Eq. 5). To depict the ideas, a trivariate case (i.e.,
$\chi = 2$) is considered here. Based on the streamflow values at the donor sites ($q_2, q_3$), $\widehat{q_1}$ can
be obtained using the conditional distribution function $h(q_1|q_2, q_3)$. For some fixed
probabilities $\phi$ (e.g., $\phi = 0.1, \dots, 0.9$), $F_1(\widehat{q_1})$ is derived from $C_{1|2,3}$ using an explicit
function:

$$C_{1|2,3}^{-1}\big(\phi|F_2(q_2), F_3(q_3)\big) = h_{1|2}^{-1}(h_{1|32}^{-1}(\phi|h_{2|1}(F_2(q_2)|F_1(q_1)))|F_1(q_1))$$    Eq. (7)




where $C_{1|2,3}^{-1}$ is the inverse of the copula function given the $\phi$ quantile curve of the copula
(Liu et al., 2015; Xu and Childs, 2013). Therefore, the $\phi$th copula-based conditional quantile
function of streamflow at the target site can be calculated as follows:

$$q_1(\phi|q_2 q_3) = F_1^{-1}(C_{1|2,3}^{-1}(\phi|F_2(q_2), F_3(q_3))) =$$
$F_1^{-1}(h_{1|2}^{-1}(h_{1|32}^{-1}(\phi|h_{2|1}(F_2(q_2)|F_1(q_1)))|F_1(q_1)))$                    Eq. (8)

Similarly, for the $k$-dimensional case, the $\phi$th copula-based conditional quantile function can
be calculated along with streamflow at the $k$-1 donor sites. To acquire an estimate at the target
site, 1000 samples from uniform distribution over the interval [0, 1] are generated using Monte
Carlo simulations. In this study, the mean value of these generations is regarded as the best
estimate.

**3. Application**
This study first explores the performance of $\mathcal{M}_{\mathrm{Dvine}}$ under synthetic example. In this analysis,
$\mathcal{M}_{\mathrm{Bicop}}$ and $\mathcal{M}_{\mathrm{Kraus}}$ are also employed to show the usefulness of $\mathcal{M}_{\mathrm{Dvine}}$. For $\mathcal{M}_{\mathrm{Bicop}}$, the
optimal bivariate copula is selected based on the AIC while the five bivariate copulas (Gaussian,
Student-t, Frank, Gumbel, and Clayton copulas) are considered as its potential candidates. A
brief description of two additional models are presented in the supporting information. After


that, those three models are used for a real application to 54 stream gages located in a region
of the eastern United States by estimating streamflow in partially gaged locations. Finally,
seven infilling approaches (Table 1) are also utilized and evaluated in a cross-validated
framework to evaluate the performance of the proposed model.

*3.1 Synthetic simulation*
Synthetic streamflow data are generated using controlled Monte Carlo experiment to explore
how well the three copula-based models ($\mathcal{M}_{\text{Bicop}}$, $\mathcal{M}_{\text{Kraus}}$, $\mathcal{M}_{\text{Dvine}}$) provide streamflow
predictions at the target site given a complex streamflow data in a pseudo gage network. In this
analysis, a six-dimensional streamflow set ($q_1^\zeta$, $q_2^\zeta$, $q_3^\zeta$, $q_4^\zeta$, $q_5^\zeta$, $q_6^\zeta$), $\zeta = 1, \dots, \xi =$
2190 (i.e. $\frac{2190}{365} = 6$ years), is modelled using four bivariate copulas (Gaussian, Student-t,
Flank, and Clayton copulas) and lognormal distributions for margins (see Figure 2).

The performance of each model is evaluated in a calibration-validation framework. First,
synthetic streamflow data are generated for six-dimensional gage network. Then, $\varphi$ years of
data are randomly selected to be assumed known at the target gage, and the streamflow for the
remaining 6-$\varphi$ years of data is then estimated as missing values ($\varphi = 4$ in this analysis). This
process is repeated 20 times to build an ensemble prediction. In particular, this study assumes
the fifth streamflow data (i.e., $q_5$) to be predicted. In this assessment, two characteristics are
considered to compare the three models: model prediction reliability and uncertainty
quantification skill. Model prediction reliability is tested using the root mean squared error
(RMSE; Eq. 6) and Nash-Sutcliffe efficiency (NSE), which are further described in Section 3.4.



Uncertainty quantification skill is judged by the ability of each model to build prediction
intervals (PIs) that correctly bound predictions (see Section 3.4). Here, coverage probabilities,
defined as the proportion of the time that true values occur into these PIs, are employed to show
the usefulness of the proposed model.

*3.2 Application to the Yadkin-Pee Dee River*
The Yadkin-Pee Dee River Basin (Figure 3), covering around 18,700 km$^2$ and one of the largest
river basins in North Carolina and South Carolina (Fisk, 2010), is used as real data to evaluate
infilling ability. The basin flows from the northwestern corner of North Carolina near Blowing
Rock and extends south by southeast, crossing the south-central border of North Carolina into
South Carolina, with slightly more than half of its watershed in North Carolina. Most of the
land covered within the basin is forested or used for agriculture although urban areas of the
basin are expanding.

Daily streamflow data at 54 gages are gathered throughout the study region from web interface
of the U.S. Geological Survey (USGS) National Water Information System (NWIS) (U.S.
Geological Survey, 2018). The 54 gages are selected based on the following criteria: (1) all
gages are recorded continuously for 15 years of daily streamflow over the period from January
2004 to December 2018, and (2) gages have non-zero daily values for the period in the first
criterion since gages with streamflow values equal to zero require a more flexible modeling
structure. Thus, it is common to model zero flows separately in regionalization studies. Based
on the second criterion, 10 gages are discarded (not shown).






*3.3 Intermodel comparison framework*

A set of seven infilling approaches is used in the final assessment (see Table 1): (1) $\mathcal{M}_{\text{FDC-IDW}}$,
(2) $\mathcal{M}_{\text{IDW-streamflow}}$, (3) $\mathcal{M}_{\text{Rho-streamflow}}$, (4) $\mathcal{M}_{\text{FDC-highestrho}}$, (5) $\mathcal{M}_{\text{DAR-streamflow}}$, (6)
$\mathcal{M}_{\text{Kriging-streamflow}}$, and (7) $\mathcal{M}_{\text{Dvine}}$. This set of seven models is tested in a cross-validation
framework under two different cases. The two cases consider situations wherein $\varphi$ have
values of 2 and 8 to represent relatively deficit- and sufficient-records for the target site. Similar
to the comparative assessment to show the usefulness of the proposed copula-based model (see
Section 3.1), each case is repeated 20 times by randomly selecting $\varphi$ years over the applied
period. The reliability of each model is evaluated using RMSE and NSE metrics over the
validated four-year period randomly selected in the remaining data (i.e., 4 years in 15-$\varphi$ years).

*3.4 Error metrics and error decomposition*

As presented in Sections 3.1 and 3.3, the root mean squared error (RMSE; Eq. 6) and Nash-
Sutcliffe efficiency (NSE) are employed to evaluate prediction skills:

$$NSE = 1 - \frac{\sum_{\zeta=1}^{\xi}(\widehat{q^{\zeta}}-q^{\zeta})^2}{\sum_{\zeta=1}^{\xi}(q^{\zeta}-\overline{q^{\zeta}})^2} \qquad \text{Eq. (9)}$$


The NSE (RMSE) can range from $-\infty$ to 1 (0 to $\infty$), with higher NSE (lower RMSE) implying
better performance. Both metrics have been commonly used in hydrology analysis (Boyle et





al., 2000).

Following derivations suggested in Gupta et al. (2009), the RMSE can be further decomposed
into three components:

$$RMSE^2 = MSE = (\hat{\mu} - \mu)^2 + (\hat{\sigma} - \sigma)^2 + 2\sigma\hat{\sigma}(1 - r)$$  Eq. (10)

where $\mu$ ($\hat{\mu}$) and $\sigma$ ($\hat{\sigma}$) represent the average and standard deviation for the observed
(estimated) streamflow, respectively, and $r$ indicates the estimated correlation coefficient.
The first component $(\hat{\mu} - \mu)^2$ is a measure of how well the average of the observed
streamflow represents the average of the estimated streamflow; the second component
$(\hat{\sigma} - \sigma)^2$ is a measure of how well the variance of the prediction represents the variance of the
observed streamflow; and the third component $2\sigma\hat{\sigma}(1 - r)$ is dominated by the correlation
and is defined as the "timing" component (Worland et al., 2019). Using these three defined
components, their absolute contributions are explored in this study.

In addition, the accuracy of the uncertainty quantification skill is also evaluated for the copula-
based models ($\mathcal{M}_{Bicop}$, $\mathcal{M}_{Kraus}$, $\mathcal{M}_{Dvine}$). To be specific, this study utilizes the PI coverage
probability (PICP), which a common metric for this purpose (He et al., 2017; Niemierko et al.,
2019). It provides the relative number of data points that fall between the defined bounds as
expressed follows:






$$\text{PICP} = \frac{1}{\xi}\sum_{\zeta=1}^{\xi}\Theta_\zeta \ \text{with} \ \Theta_\zeta = \begin{cases} 1, & if \quad q^\zeta \in [L^\zeta, U^\zeta] \\ 0, & else \end{cases} \qquad \text{Eq. (11)}$$


where $\Theta_\zeta$ is the indicator variable if $q^\zeta$ is covered by the $\zeta$th PI defined by the lower bound

$L^\zeta$ and upper bound $U^\zeta$. This study examines the prediction accuracy of single quantiles.

Therefore, the lower bound is defined as $L^\zeta = -\infty$ and $U^\zeta = \widehat{q^{\zeta,\varpi}}$ where $\varpi$ is the

estimated quantile at time $\zeta$. Accordingly, the upper bound is not a constant, but is re-assigned.

By subtracting the nominal confidence $\varpi$ from PICP, the average coverage error (ACE) is

obtained as follows:

$$\text{ACE} = \text{PICP} - \varpi \qquad \text{Eq. (12)}$$

The metric clearly indicates if the predicted quantile is underestimated (ACE < 0) or

overestimated (ACE > 0) while taking small values around 0 for ideal case.

**4. Results**

*4.1 Results for synthetic experiment*

Prediction results from out-of-samples for the RMSE and NSE metrics are presented for the

three copula-based models ($\mathcal{M}_{\text{Bicop}}$, $\mathcal{M}_{\text{Kraus}}$, $\mathcal{M}_{\text{Dvine}}$) in Table 2. The ACE scores are also



described for $\varpi \in \{0.05, 0.10, 0.50, 0.90, 0.95\}$ in Table 3. When compared to the other
models, $\mathcal{M}_{\text{Bicop}}$ achieves lower RMSE values in the right tail of the RMSE distribution over
the validation periods, but severely underperforms the majority of the designed experiment,
suggesting this model formulation relying on a single donor leads to poor predictions. $\mathcal{M}_{\text{Kraus}}$
provides higher RMSE values for all the RMSE distribution, particularly for the right tail of
the RMSE distribution. The model utilizes streamflow data from all donors (i.e., five donor
sites) although the first two gages (Gages 1 and 2) show insignificant associations to the target
site ($r_{1,5} = 0.11$ and $r_{2,5} = 0.14$). $\mathcal{M}_{\text{Dvine}}$ unequivocally produces the best predictions.
$\mathcal{M}_{\text{Dvine}}$ adopts streamflow data from two or three donors (Gages 3, 4 and 6) without utilizing
streamflow data from the first two donors when a multiple dependence structure is established
to build an ensemble prediction. It outperforms $\mathcal{M}_{\text{Bicop}}$ and $\mathcal{M}_{\text{Kraus}}$ across all validation
periods, besides a few with the worst performance. Even in this case, the maximum RMSE of
$\mathcal{M}_{\text{Dvine}}$ is fairly less than the maximum RMSE of $\mathcal{M}_{\text{Kraus}}$.

In addition, the ACE results present how the three models characterize prediction uncertainty.
$\mathcal{M}_{\text{Dvine}}$ is capable of properly covering the predications across the entire distribution while
slight overestimation occurs for the smallest two quantiles. The remaining upper quantiles also
tend to slightly overestimate the true values but the overestimations are less than the other
models ($\mathcal{M}_{\text{Bicop}}$, $\mathcal{M}_{\text{Kraus}}$). Taken together, the results of the synthetic experiment suggest that
$\mathcal{M}_{\text{Dvine}}$ yields the best predictions among the copula-based models tested.

*4.2 Performance of the copula-based models in the Yadkin-Pee Dee River*





Using the insights developed from the synthetic experiment above, the three copula-based
models are applied to the streamflow data for the Yadkin-Pee Dee River. At first, upper and
lower tail dependences ($\lambda_{upper}$ and $\lambda_{lower}$) are examined for all two pairs of sites (see Figure
4) using the approach of Schmid and Schmidt (2007). Theoretical background is described in
the Supporting Information (Text S3). Note that in this analysis, the dependences become more
obvious as the values approach unity. Two major insights emerge from this figure. First, many
site-pairs exhibit strong upper tail dependence, suggesting that streamflow variability has a
tendency to be more correlated under high-flow conditions compared to under low-flow
conditions (i.e., asymmetric dependence). The lack of lower-tail dependence may be due to
contributions governing low streamflow such as river regulation. Next, even under high- or
low-flow conditions, there is a wide range of tail dependence across the study basin (i.e.,
heterogeneous dependence). To sum up, a wide range of complex dependencies is observed in
the streamflow data over the study basin. The complex dependences suggest, when streamflow
is estimated from multiple donors, the potential usefulness of considering a multiple
dependence structure, which is one of the main features of vine copulas.

Figure 5 shows the RMSE and NSE results for the three copula-based models under a "leave-
one-out" cross validation framework. This process is repeated 20 times to build an ensemble
prediction by using test periods randomly defined. For this analysis, five years of data are
selected to be assumed as the observed period at the target gage, and another four years are
randomly selected in the remaining data for the test period. Similar to the results from the
synthetic experiment, $\mathcal{M}_{Kraus}$ performs poorly in both the RMSE and NSE metrics (median
RMSE = 1.549 and NSE = 0.652). The bivariate copula performs well (median RMSE =



1.496), indicating that this approach efficiently leverages available information even though
the information is limited to single donor. Particularly, $\mathcal{M}_{\text{Bicop}}$ achieves lowest RMSE values
in the upper side of the RMSE box (e.g., third quartile), providing a strong uncertainty
quantification skill for the upper bound. However, $\mathcal{M}_{\text{Dvine}}$ yields the best median RMSE and
NSE values (= 1.359 and 0.719). Given the heterogeneous dependence conditions (see Figure
4), the high dimensional structures are effective in modeling a complex streamflow gage
network. This feature can substantially improve prediction of target site flows.

Figure 6a presents the ACE scores described for principal quantiles, $\varpi \in$
$\{0.05, 0.10, 0.20, \ldots, 0.90, 0.95\}$, across all target sites under the cross validation framework.
Figure 6b presents 95% PIs for each model for an example time period (1 May 2018 to 31 July
2018) for one target site (USGS site ID: 02143500). Note that the ACE would ideally take zero
value, regardless of the quantiles. The ACE scores for the three models ($\mathcal{M}_{\text{Bicop}}$, $\mathcal{M}_{\text{Kraus}}$,
$\mathcal{M}_{\text{Dvine}}$) range from 0.004 to 0.0007 when considering all the quantiles together. However, the
scores vary depending on the quantiles. For instance, the ACE score for $\mathcal{M}_{\text{Kraus}}$ is noticeably
positive but is almost zero around the median streamflow, indicating that the model properly
represent uncertainty of the median streamflow. $\mathcal{M}_{\text{Bicop}}$ and $\mathcal{M}_{\text{Dvine}}$ result in very similar
ACE scores although $\mathcal{M}_{\text{Dvine}}$ performs slightly better than $\mathcal{M}_{\text{Bicop}}$. The differences in
characterization of prediction uncertainty can be confirmed from a particular target site (Figure
6b).

Based on the results in Figures 5 and 6, $\mathcal{M}_{\text{Dvine}}$ has the most reliable overall performance (as





judged by model prediction reliability and uncertainty quantification skill) and is selected as
an appropriate copula model to infill missing data in partially gaged. Figure 7 shows an
example application of $\mathcal{M}_{\mathrm{Dvine}}$ including the optimal donor sites, proper bivariate copulas
and their parameters for one target site (USGS site number #214645022) when the model is
calibrated using the full 15-year record.

*4.3 Intermodel comparison for streamflow estimation*
To assess the predictive skill of the proposed vine copula model, it is compared with six other
statistical models (see Table 1). Figure 8 shows RMSE and NSE for the seven models where
the streamflow values are estimated based on the available data defined by the two different
cases, labeled "deficit record" and "sufficient record" (see Section 3.3). Under all cases, the
vine copula approach outperforms the other infilling approaches. For example, for the
"sufficient record" case, median NSE for $\mathcal{M}_{\mathrm{Dvine}}$ is 0.673 whereas those for
$\mathcal{M}_{\mathrm{IDW-streamflow}}$ and $\mathcal{M}_{\mathrm{rho-streamflow}}$ are 0.462 and 0.649, respectively. In this analysis, the
approaches, which are based on streamflow values of the donor sites without utilizing non-
exceedance probability including DAR-streamflow and Kriging-streamflow, yield relatively
increased bias in their predictions. On the other hand, an application of FDC models offers
reliable predictions. For instance, for the "sufficient record" case, median RMSE for
$\mathcal{M}_{\mathrm{FDC-highestrho}}$ is 1.603 compared to that of a direct of using streamflow (e.g., median RMSE
of $\mathcal{M}_{\mathrm{FDC-streamflow}}$ = 3.422 for the sufficient record). Similar interpretation can be found in
the comparison between $\mathcal{M}_{\mathrm{FDC-IDW}}$ and $\mathcal{M}_{\mathrm{IDW-streamflow}}$. The results from these approaches
suggest that utilizing FDC process leads to a reliable estimation, which is a primary structure
in the vine copula. The other noticeable feature is that available data length provides a



significant influence on performance of some infilling methods. In particular, this is quite
evident for the vine copula model (median RMSEs: 1.598 and 1.379 for deficit and sufficient
records, respectively).

*4.4 Prediction error decomposition*
The RMSE is decomposed into their components (bias, variance, and timing components) for
both the "deficit record" and "sufficient record" predictions (Figure 9). For the both cases,
prediction errors for all seven models are caused largely by timing components. In particular,
models estimating directly streamflow values (IDW-streamflow, DAR-streamflow, Kriging-
streamflow) produce a somewhat biased component, which increases when a shorter record is
employed in the model. For instance, the timing component for $\mathcal{M}_{\text{IDW-streamflow}}$ is 4.11 and
3.75 for the "deficit record" and "sufficient record", respectively. Moreover, timing
components dominate the error metric for all cases. However, the importance of variance
component is increased, especially in three models (FDC-IDW, DAR-streamflow, Kriging-
streamflow). Lastly, the results inform that if the proposed vine copulas approach is adapted,
variance and timing components are better captured, leading to better streamflow estimations,
which is beneficial in the practical applications of water resources management.

Finally, two predictions are further produced using two additional experiments: (1) the
observed marginal cumulative probabilities (i.e., using all 15 years) and conditional streamflow
values constructed from the partial record (i.e., based on $\varphi$ years), and (2) the estimated
marginal cumulative probabilities (i.e., based on $\varphi$ years) and conditional streamflow values



constructed from the full record (i.e., all 15 years). Their prediction abilities are evaluated over
the validated four-year period randomly selected in the remaining data. Similar to the previous
analysis, each analysis is tested 20 times. The results from these experiments provide an
inference to better isolate how error components from the two-step procedure (see section 2.2)
influence prediction skill.

Figure 10 shows the ACE scores from the out-of-sampled predictions using the proposed Dvine
model under the two scenarios. When considering all the quantiles together, the ACE scores
for the two scenarios are 0.003 (scenario #1) and 0.006 (scenario #2) on average under the
"deficit record" prediction. Also, the scores under the "sufficient record" prediction are all
nearly 0.003. Those results of the scores are sufficiently closed to zero, implying that both
predictions are reliable. Yet, compared to the predictions estimated by the cumulative
probabilities estimated by the partial record, and conditional models constructed by full records
(i.e., scenario #2), the ACE scores are achieved better, if the cumulative probabilities are
determined by the full record, except for some of the low and high quantiles. Similar
interpretation can be found in the NSE performance of two scenarios (see insets of Figure 10).
It may suggest that the first procedure (i.e., how to determine the cumulative probabilities for
the target site and its donors) is needed to pay careful attention when $\mathcal{M}_{\mathrm{Dvine}}$ is utilized.
Nevertheless, the procedure to construct the conditional model in a streamflow gage network
is obviosuly crucial since the over or under-estimations are observed in many quantiles when
the insufficient sampling is employed in this process.

**5. Conclusion**





This study introduces a multiple dependence conditional model (i.e., vine copulas) to produce
streamflow estimates at partially gaged sites. The model includes a flexible high dimensional
joint dependence structure and conditional bivariate copula simulations. In order to confirm the
usefulness of a multiple dependence structure and the procedure for an appropriate number of
donor sites in the final vine copula model, the bivariate copula model and two types of vine
copulas with their unique procedure to determine the optimal number of donor sites are first
investigated using the generated data. These analyses were further extended in a case study of
the Yadkin-Pee Dee River Basin, the eastern United States by estimating streamflow in partially
gaged locations. In this analysis, six statistical infilling approaches were also employed to
represent applicability of the proposed model.

Results of the synthetic experiment and application to the Yadkin-Pee Dee River Basin
demonstrate that the propose model has benefits in some aspects. First, a multiple dependence
structure adopted in the proposed model is beneficial. From the massive evaluation experiments,
this study shows that multiple dependence structure clearly outperforms a single dependence
structure although there is the risk of overfitting when too many dependence structures are
employed. Moreover, this study confirms that the proposed multiple dependence structure
model with their optimum number of donor sites produces more reliable streamflow estimation
than other common infilling models. Next, the proposed model allows the development of
confidence intervals to consider prediction uncertainty, which is fairly attractive compared to
other models. For example, Bárdossy and Pegram (2013) argue that confidence intervals
obtained using an ordinary kriging model do not reflect the prediction uncertainty well
particularly on a daily scale. Overall, this study exhibits that a vine copula is potentially an
effective tool to support water resource management planners for objectives like gap-filling or





extending missing streamflow records.

While the results of the proposed model are favorable, there are possible limitations worthy of
further discussion. First, the assessment illustrated in this study focuses on model performance
under cross-validation at partially gaged basins, but additional work is needed to extend the
proposed model to ungagged basins. One possible way is to build a regression based model
with spatial proximity and physical basin characteristics to define associations between the
target and donor sites (e.g., Ahn and Steinschneider, 2019). Second, this study does not
consider potential nonstationarity in FDCs and correlations caused by the influence of
anthropogenic activity and change in land use. Nonstationarity may not be problematic in this
analysis since the assessment is limited to 15 years across the gaging network. However, if
longer records were used, it would be beneficial to consider the potential nonstationarity. This
exploration is left for future work.

There are several opportunities to improve the model structure. For instance, a vine copula is
able to incorporate more additional conditioning variables. One feasible approach is to add a
time series of climate data (e.g., precipitation) or to decompose a time series of streamflow
from the donor sites into a number of periodic components at different frequency levels through
the wavelet decomposition approach (Kisi and Cimen, 2011).

Lastly, the results presented here are specific to a study basin used in a case study. The proposed
model has not restricted to other watersheds around the world and its application is further


required towards drawing more generalized conclusions. In addition, the model could be used
for the purpose of infilling missing values of other hydrometeorological variables besides
streamflow (e.g., precipitation and soil moisture). For this application, the implementation of a
vine copula with combined discrete and continuous margins (i.e., to account for no rainfall
days) should be explored (e.g., Stoeber et al., 2013).

**Acknowledgements**

This work was supported by the National Research Foundation of Korea (NRF) grant funded
by the Korea government (MSIT) (No. 2019R1C1C1002438). Also, the author would like to
acknowledge Scott Steinschneider for his helpful comments during the development of this
paper.

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




**List of Figures**






**List of Tables**
Table 1 Seven infilling approaches discussed in the study.

Table 2 RMSE and NSE results over the validation periods under synthetic experiment for
comparing copula-based model formulations. Best metric values for each quantile
are italicized and bolded.

Table 3  Results of average coverage error (ACE) over the validation periods under
synthetic experiment for comparing copula-based model formulations. Best
metric values for each quantile are italicized and bolded.







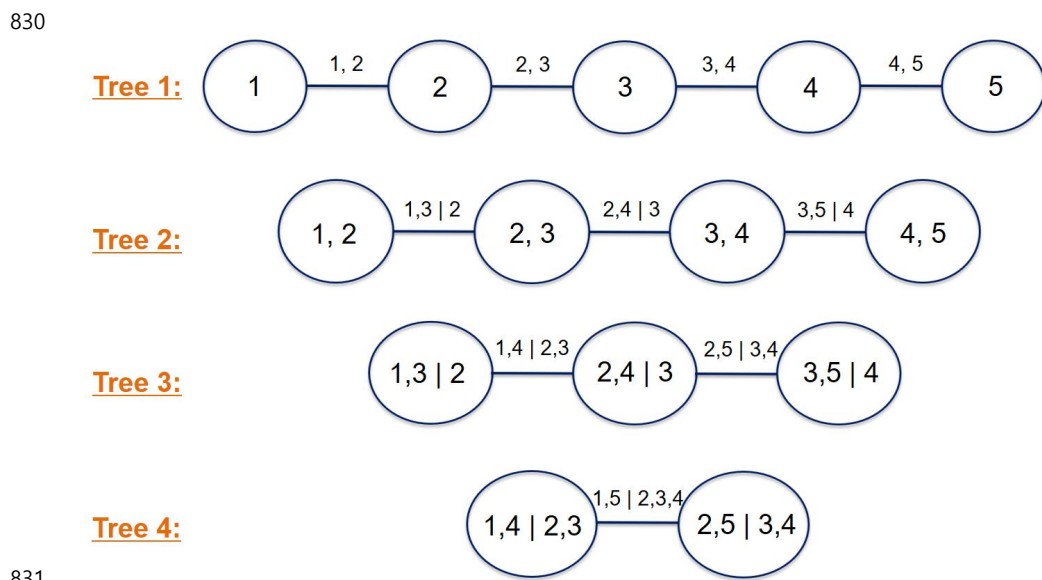

Figure 1 Example of D-vine structures with 5 variables, 4 trees and 10 edges






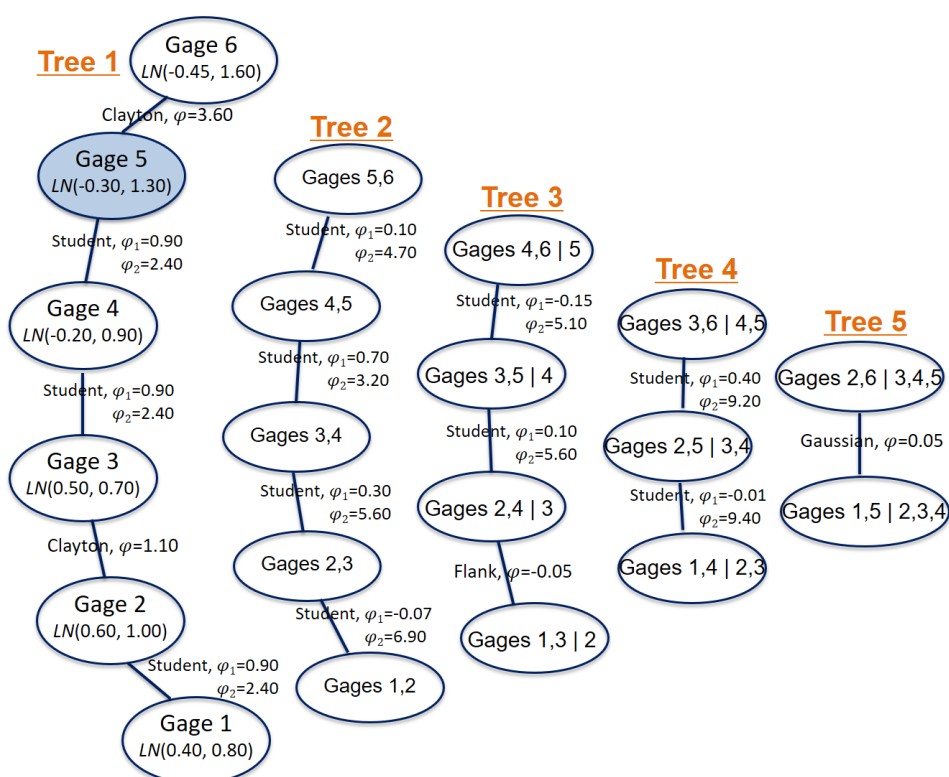

Figure 2 Structure of the 6-Dimensional vine model and marginal for the synthetic simulation.
$LN(\pi, \sigma^2)$ denotes the log normal distribution with its mean $(\pi)$ and variance $(\sigma^2)$. The target
gage is highlighted.






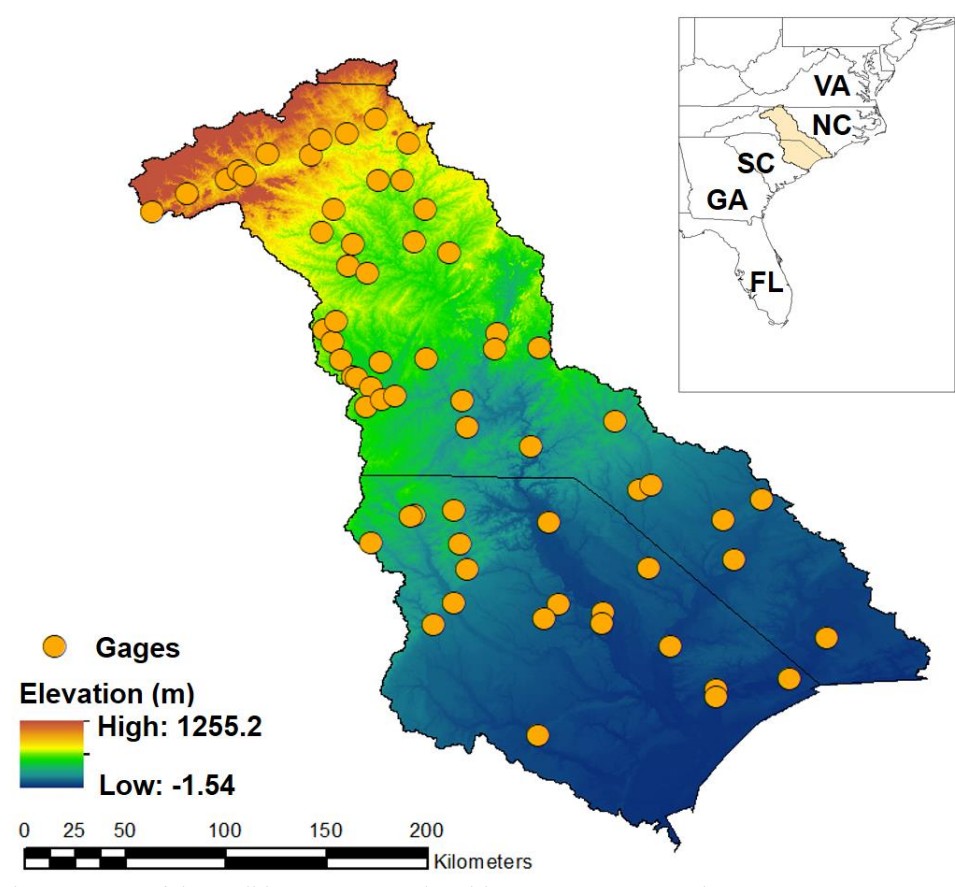

Figure 3 Map of the Yadkin-Pee Dee Basin with 54 stream gage stations
.


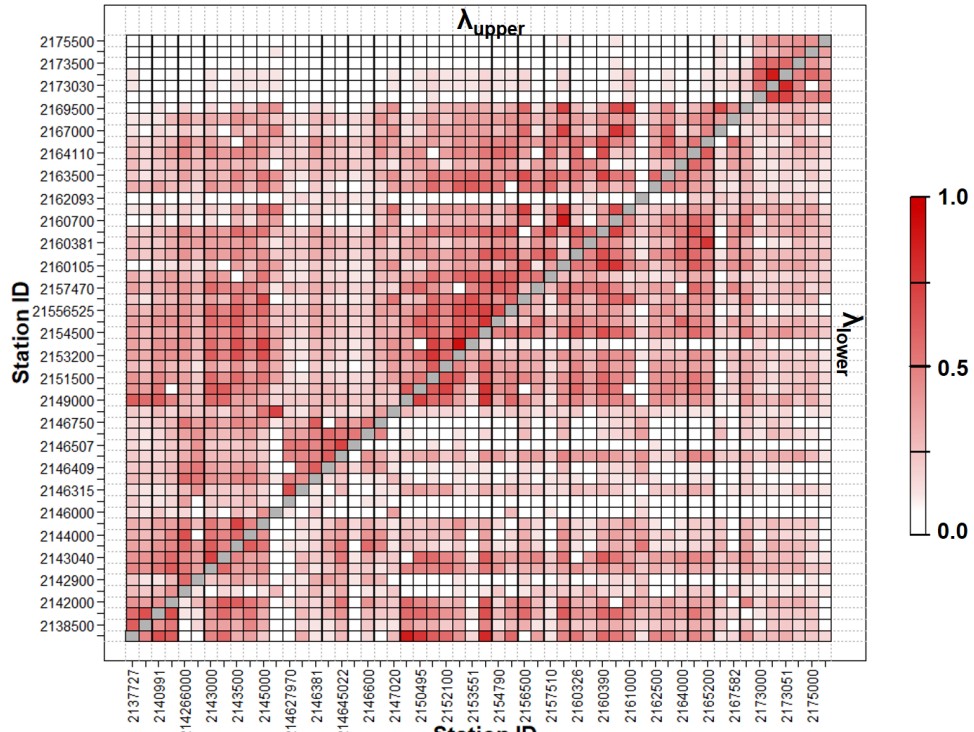

Figure 3 Density of the Akaike Information Criterion (AIC) for the six univariate
distributions applied across sites in the study basin. Mean values for each AIC
density are also presented.

Figure 4 Pairwise upper and lower tail dependence for watersheds in the Yadkin-Pee Dee River
Basin. The upper triangular matrix shows values for the upper-tail dependence and the lower
triangular matrix presents values for the lower-tail dependence.






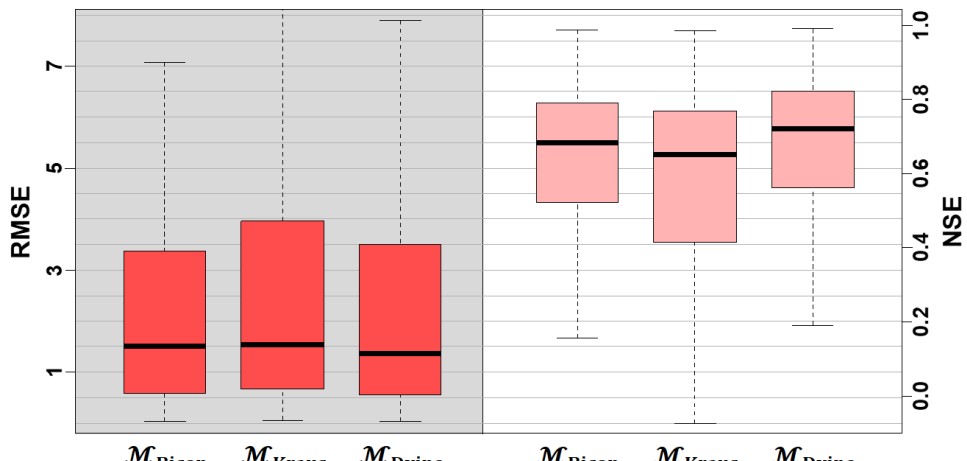

Figure 5 Model performance for the Yadkin-Pee Dee river under a cross-validation framework
based on RMSE (dark squares) and NSE (light squares).






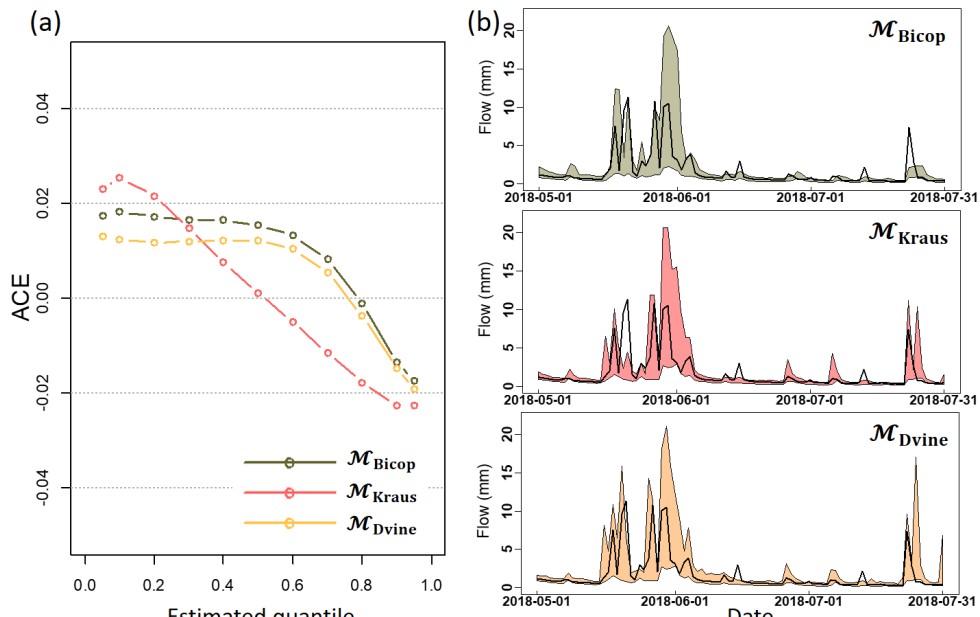

Figure 6 (a) Average coverage error from three copula-based models for the Yadkin-Pee Dee
River Basin across exemplarily quantiles, and (b) 95% PIs for three models for an example
period (1 May 2018 to 31 July 2018) for a specific target gauge (USGS ID: 02143500).
Observed streamflow (black solid line) is also presented in each figure.




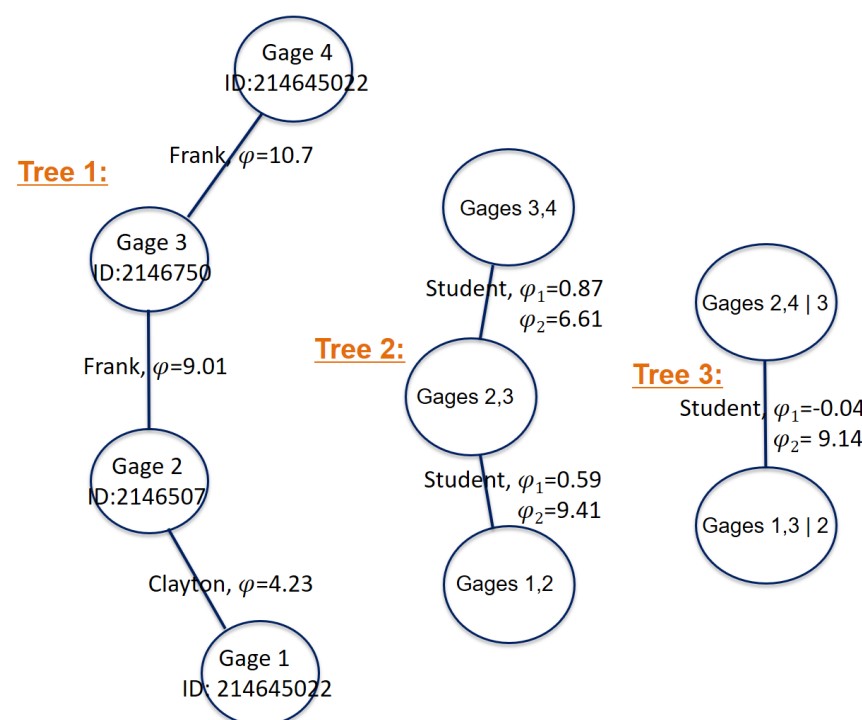

Figure 7 Structure of the Dvine copula applied for a particular target site (USGS site number
214645022) with the defined bivariate copulas and their parameters.


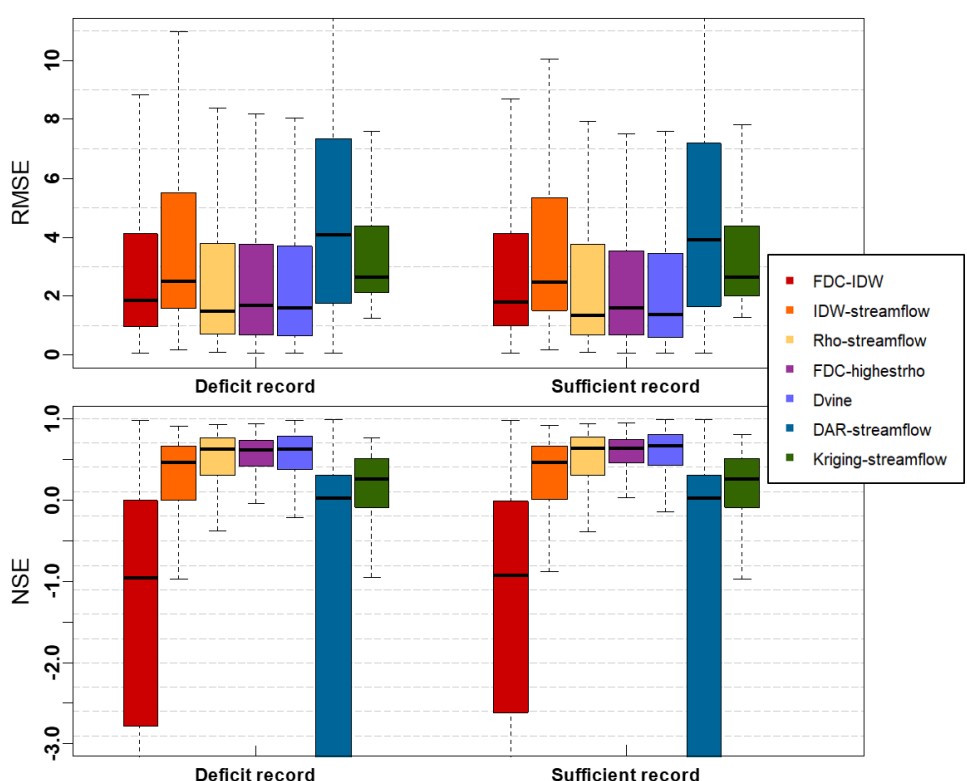

Figure 8 Inter-model comparison using cross-validation experiments based on RMSE (upper)
and NSE (lower).






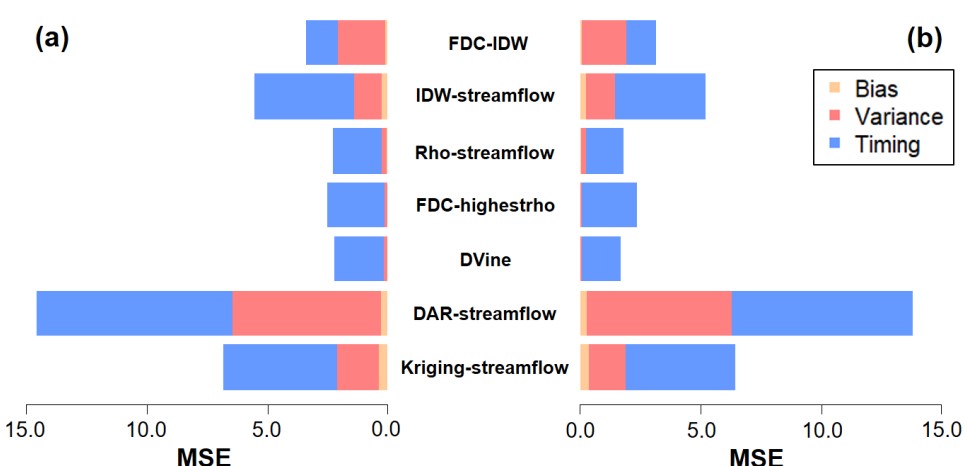


Figure 9 Three contributions from the decomposed mean squared error (MSE) for the cross-
validation experiment with (a) the deficit record and (b) sufficient record scenarios.


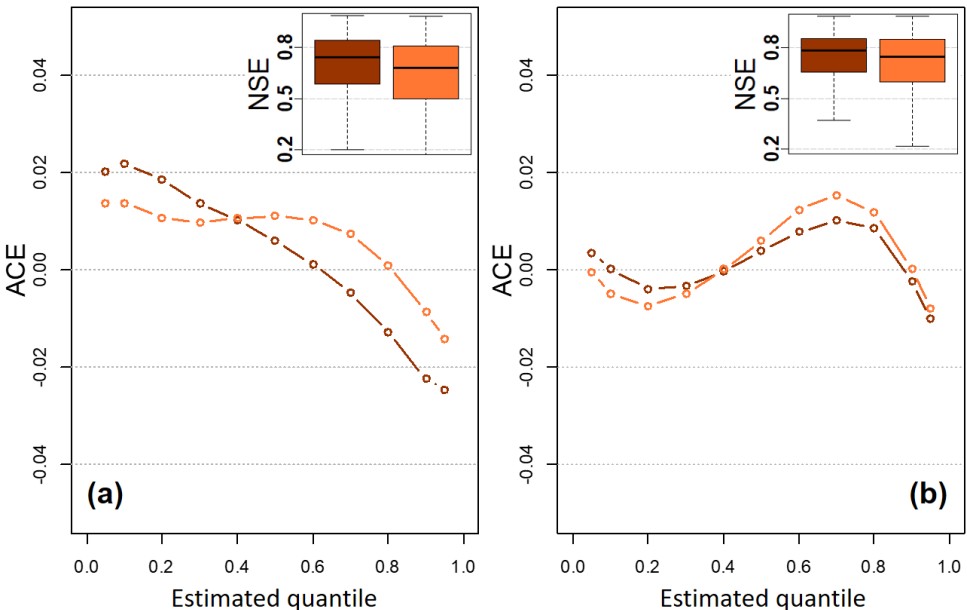

Figure 10 Average coverage error of the Dvine model for two scenarios under (a) the "deficit"
and (b) "sufficient" cases. In each case, the dark line represents the scenario by the marginal
cumulative probabilities using all years and conditional streamflow values constructed from
the partial record. On the other hand, the light line illustrates the scenario by the marginal
cumulative probabilities estimated by the partial record and conditional streamflow values
constructed from the full record. Inset: NSE performance of the Dvine model for the two
scenarios in each case.





Table 1 Seven infilling approaches discussed in the study

| No. | Method | Description |
| --- | --- | --- |
| 1 | FDC-IDW | Inverse distance-weighted estimate of non-exceedance probability from those of all donors. |
| 2 | IDW-streamflow | Inverse distance-weighted estimate using streamflow from all donors. |
| 3 | Rho-streamflow | Correlation-weighted streamflow estimate from the selected donors for each time step. The optimal number of donors is determined in a cross-validation framework. |
| 4 | FDC-highestrho | Estimate non-exceedance probability from the gage with the highest correlation. |
| 5 | DAR-streamflow | Drainage-area (DA) ratio for streamflow using the DA from the nearest neighbor gage. |
| 6 | Kriging-streamflow | Geostatistical interpolation method to estimate streamflow from all donors for each time step. |
| 7 | DVine | Vine copula-based estimate from the selected donors |





Table 2 RMSE and NSE results over the validation periods under synthetic experiment for
comparing copula-based model formulations. Best metric values for each quantile are
italicized and bolded.

| Metric | Model formulation | Min | First quantile | Median | Third quantile | Max |
|--------|-------------------|-----|----------------|--------|----------------|-----|
| Root mean squared error (RMSE) | $\mathcal{M}_{Bicop}$ | 0.912 | 1.119 | 1.258 | 1.363 | ***3.353*** |
| | $\mathcal{M}_{Kraus}$ | 0.990 | 1.140 | 1.386 | 1.660 | 4.273 |
| | $\mathcal{M}_{Dvine}$ | ***0.895*** | ***1.046*** | ***1.112*** | ***1.391*** | 4.119 |
| Nash-Sutcliffe efficiency (NSE) | $\mathcal{M}_{Bicop}$ | ***0.464*** | 0.779 | 0.826 | 0.856 | 0.902 |
| | $\mathcal{M}_{Kraus}$ | 0.198 | 0.724 | 0.782 | 0.825 | 0.885 |
| | $\mathcal{M}_{Dvine}$ | 0.248 | ***0.805*** | ***0.838*** | ***0.869*** | ***0.905*** |





Table 3 Results of average coverage error (ACE) over the validation periods under synthetic
experiment for comparing copula-based model formulations. Best metric values for
each quantile are italicized and bolded.

| Model formulation | Estimated quantile ($\varpi$) | | | | |
| :---: | :---: | :---: | :---: | :---: | :---: |
| | 0.05 | 0.10 | 0.50 | 0.90 | 0.95 |
| $\mathcal{M}_{\text{Bicop}}$ | 0.027 | 0.063 | 0.079 | 0.014 | 0.002 |
| $\mathcal{M}_{\text{Kraus}}$ | ***0.003*** | ***0.011*** | 0.055 | 0.024 | 0.001 |
| $\mathcal{M}_{\text{Dvine}}$ | 0.029 | 0.048 | ***0.042*** | ***0.001*** | ***0.000*** |

