# Peer review of "Streamflow estimation at partially gaged sites using multiple dependence conditions via vine copulas Kuk-Hyun Ahn1 1Assistant Professor, Department of Civil and Environmental Engineering, Kongju National University, Cheon-an, South Korea; Corresponding author; e-mail: ahnkukhyu"

_Hydrology and Earth System Sciences, 2020_

## Referee Comment (RC1) · Anonymous Referee #1 · 5 Jan 2021

Dear editor,

In this study several models were used to fill missing data in the field of hydrology. The present study is a very timely study and fits into the scope of the Journal. The paper is quite informative because it provides information on the to infill missing data. Although there are different interpolation methods and different models such as time series, black box models as well as hybrid models that may provide better results. Also, in the use of interpolation models, the accuracy of the simulations can be increased by using copula-based models. However, this manuscript is well prepared according to the intended methods. Considering the used models, the present manuscript is accepted to publish. Several suggestions have been made to improve the status of the manuscript. In the abstract, it is better to present the improvement percentage of

the superior model compared to other models Choose keywords that do not match the title of the manuscript. The conclusion section should be more concise and the main results should be presented.

---

## Referee Comment (RC2) · Anonymous Referee #2 · 18 Mar 2021

**Summary**

This work presents a copula-based approach to estimate streamflow at partially gaged stations. The author describes the theoretical framework and tests the new methodology on a simulation study and a case study of gages located within the Yadkin-Pee Dee River Basin in the eastern United States. Besides, the author carries out a throughout comparison between the new approach and other available methods for infilling.

**Main Points**

The paper is well written and informative, and it is of great interest to the general audience of the HESS journal. I have two main points to discuss with the author.

1. The author compares the Vine-copula approach with 1) a bivariate copula model and 2) another type of vine copula approach based on a different choice of co-variates. From the results presented in the paper, I am under the impression that the bivariate copula model performs reasonably well. What is the computational burden of using a Vine copula structure instead of the simpler bivariate copula model? Could the author add a few lines on the computational aspects of the newly introduced method?

2. Looking at the results of Figure 8, I notice that the *FDC-highestrho* and *DVine* approaches seem to be very close in terms of their performance. This result makes it a bit unclear to me the role and contribution of the pair-wise dependences (e.g., Gumbel versus Clayton) in the methodology. Perhaps a neglected aspect of the analysis is the robustness of the results under misspecifications of the chosen pair-wise copulas with the same pair-wise Kendall's tau assumptions. I would appreciate it if the author could elaborate on this point.

**Minor points**

Line 101: I do not understand the meaning of the word "efficient" in this context.

Line 118: Citing the most recent book by Joe "Dependence modeling with copu-las" (2014) would be more appropriate.

Line 123: I do not understand this sentence.

Line 246: "Penalized" instead of "panelized".

Line 905: Improve the readability of the figure (the plot labels cover the matrix).

Line 929: Specify how to interpret the scores in the figure caption (i.e., the higher the score, the better). The same comment applies to the other figures.

Line 990: Improve the readability of the figure (there is some text outside the circles).

---

## Author Comment (AC1) · 20 Mar 2021

**< REPLY TO REVIEWER 1>**

- **Title: Streamflow estimation at partially gaged sites using multiple dependence conditions via vine copulas**
- **Authors: Kuk-Hyun Ahn**

**((Acknowledgement))** The authors sincerely thank the reviewer for their helpful and constructive comments.

**((Comment #1))**

In the abstract, it is better to present the improvement percentage of the superior model compared to other models.

**((Reply))**

The point of the reviewer is well taken. During the revision of the paper, the improvement percentage will be provided in the abstract.

**((Comment #2))**

Choose keywords that do not match the title of the manuscript.

**((Reply))**

Keywords will be modified based on the title of the manuscript.

**((Comment #3))**

The conclusion section should be more concise and the main results should be presented.

**((Reply))**

After re-reading our previous manuscript, I realized that the conclusion did not provide enough detail about the main results to summarize the draft. More summary statement will be inserted in the conclusion.

---

## Author Comment (AC2) · 20 Mar 2021

**< REPLY TO REVIEWER 2>**

- **Title: Streamflow estimation at partially gaged sites using multiple dependence conditions via vine copulas**
- **Authors: Kuk-Hyun Ahn**

**((Acknowledgement))** The authors sincerely thank the reviewer for their helpful and constructive comments.

**((Comment #1))**

Could the author add a few lines on the computational aspects of the newly introduced method?

**((Reply #1))**

This is a good point. I admit that the proposed method is computationally expensive. Thus, this study adopts the multicore processing to reduce the computational burden. The limitation could be problematic particularly for a larger, complex streamflow gaging network. However, the computational burden will be excused since many local water managers may not need to build the model repeatedly whenever they meet missing values. Instead, they can use the model for a while once the proposed model is calibrated for a specific site. The argument will be raised in the conclusion section.

**((Comment #2))**

Perhaps a neglected aspect of the analysis is the robustness of the results under misspecifications of the chosen pair-wise copulas with the same pair-wise Kendall's tau assumptions. I would appreciate it if the author could elaborate on this point.

**((Reply #2))**

You are correct that the FDC-highestrho and DVine approaches seem to be very close in terms of their performance, although the DVine performs slightly better than the FDC-highestrho (e.g., median RMSEs: 1.598 (DVine) and 1.603 (FDC-highestrho) for the "sufficient record" case). I was not aware of the simplifying assumptions while developing this paper, and I am grateful to the reviewer for pointing this out. I will include this assumption in the conclusion section as a part of the areas that need to be considered for future work to improve the proposed model.

**((Minor comment #1))**

Line 101: I do not understand the meaning of the word "efficient" in this context.

**((Reply))**

The word will be altered to "effective".

**((Minor comment #2))**

Line 118: Citing the most recent book by Joe "Dependence modeling with copulas" (2014) would be more appropriate.

**((Reply))**

The recent book will be cited.

**((Minor comment #3))**

Line 246: "Penalized" instead of "panelized".

**((Reply))**

The word will be corrected.

**((Minor comment #4))**

Line 905: Improve the readability of the figure (the plot labels cover the matrix).

**((Reply))**

The readability of the figure will be improved following the reviewer's comment.

**((Minor comment #5))**

Line 929: Specify how to interpret the scores in the figure caption (i.e., the higher the score, the better). The same comment applies to the other figures.

**((Reply))**

The interpretation for the scores will be suggested in Figures 4, 5 and 8.

**((Minor comment #6))**

Line 990: Improve the readability of the figure (there is some text outside the circles).

**((Reply))**

The readability of the figure will be improved based on the reviewer's request.

---

## Author Response (AR1)

**< REPLY TO EDITOR>**

- **Title: Streamflow estimation at partially gaged sites using multiple dependence conditions via vine copulas**
- **Authors: Kuk-Hyun Ahn**

**((Acknowledgement))** The authors sincerely thank the editor for his helpful and constructive comments.

**((Comment #1))**

Why do you have considered these five families of Copulas: Gaussian, Student-t, Frank, Gumbel, and Clayton copulas? In literature you can find a wide variety of families of Copulas including also extreme value copulas (EVC)

**((Reply #1))**

The editor is correct that a wide variety of families can be utilized to build the copula model. As denoted in Lines 238-243, the study employs five candidates (Gaussian, Student-t, Frank, Gumbel, and Clayton copulas). I believe the five candidates are sufficient to represent diverse dependence structures. For example, a Gumbel copula may be proper for the situation wherein a higher percentile exhibit tail dependence, where high flows are connected by same rainfall events but low flows are not related. Asymmetric lower tail dependence is also possible by a Clayton copula. This situation is occasionally feasible since base flow for two basins is supplied from the same aquifer system whereas high flows are affected by local convective systems that generate for one or the other basin. In addition, those candidates have been commonly used in literature (e.g., Chen et al., 2015; Liu et al., 2015).

However, I also agree that a wide variety of families are further applicable. Therefore, I have included the possibility to consider more families of Copulas in the manuscript (Lines 236-239).

*"To build the D-vine copula model, five bivariate copulas (Gaussian, Student-t, Frank, Gumbel, and Clayton copulas) are considered as potential pair copulas (building blocks) although more families of Copulas such as extreme value copulas (EVC) are desirable. The five candidates may be sufficient to represent diverse dependence structures."*

**((Comment #2))**

Concerning the application of D-vine copulas in hydrology, I would like to point to your attention also Shafaei et al. (2017).

**((Reply #2))**

The reference that the editor suggested have been inserted as an example of hydrological application (Line 128).

**((Reference))**

Chen, L., Singh, V. P., Guo, S., Zhou, J., & Zhang, J. (2015). Copula-based method for multisite monthly and daily streamflow simulation. *Journal of Hydrology*, *528*, 369–384.

Liu, Z., Zhou, P., Chen, X., & Guan, Y. (2015). A multivariate conditional model for streamflow prediction and spatial precipitation refinement. *Journal of Geophysical Research: Atmospheres*, *120*(19), 10–116.

**< REPLY TO REVIEWER 1>**

- **Title: Streamflow estimation at partially gaged sites using multiple dependence conditions via vine copulas**
- **Authors: Kuk-Hyun Ahn**

((Acknowledgement)) The authors sincerely thank the reviewer for their helpful and constructive comments.

((Comment #1))

In the abstract, it is better to present the improvement percentage of the superior model compared to other models.

((Reply))

The point of the reviewer is well taken. The improvement percentage of the proposed model has been inserted (Line 38).

((Comment #2))

Choose keywords that do not match the title of the manuscript.

((Reply))

Based on the title of the manuscript, keywords have been modified, including vine copulas, multiple dependence condition model, infilling approach, and streamflow estimation at partially gaged site (see Line 46).

((Comment #3))

The conclusion section should be more concise and the main results should be presented.

((Reply))

Following the reviewer's recommendation, summary statement has been additionally presented in the conclusion section. For example, the improvement percentage of the proposed model is presented when it is compared to the bivariate model (Line 541). Also, the improvement for the "sufficient record" case has been suggested when the model is compared to common infilling technique (Line 545).

**< REPLY TO REVIEWER 2>**

- **Title: Streamflow estimation at partially gaged sites using multiple dependence conditions via vine copulas**
- **Authors: Kuk-Hyun Ahn**

**((Acknowledgement))** The authors sincerely thank the reviewer for their helpful and constructive comments.

**((Comment #1))**

Could the author add a few lines on the computational aspects of the newly introduced method?

**((Reply #1))**

This is a good point. I admit that the proposed method is computationally expensive. Thus, this study adopts the multicore processing to reduce the computational burden. The limitation could be problematic particularly for a larger, complex streamflow gaging network. However, the computational burden will be excused since many local water managers may not need to build the model repeatedly whenever they meet missing values. Instead, they can use the model for a while once the proposed model is calibrated for a specific site. This information has been suggested in the conclusion section (Lines 555-560).

**((Comment #2))**

Perhaps a neglected aspect of the analysis is the robustness of the results under misspecifications of the chosen pair-wise copulas with the same pair-wise Kendall's tau assumptions. I would appreciate it if the author could elaborate on this point.

**((Reply #2))**

The point is well taken. I was not aware of the simplifying assumptions while developing this paper, and I am grateful to the reviewer for pointing this out. This assumption is now included in the conclusion section as a part of the areas that need to be considered for future work to improve the proposed model (Lines 575-582).

**((Minor comment #1))**

Line 101: I do not understand the meaning of the word "efficient" in this context.

**((Reply))**

The word is altered to "effective".

**((Minor comment #2))**

Line 118: Citing the most recent book by Joe "Dependence modeling with copulas" (2014) would be more appropriate.

**((Reply))**

The recent book has been cited (see Line 118).

**((Minor comment #3))**

Line 246: "Penalized" instead of "panelized".

**((Reply))**

The word has been corrected (see Line 248).

**((Minor comment #4))**

Line 905: Improve the readability of the figure (the plot labels cover the matrix).

**((Reply))**

The readability of the figure has been improved following the reviewer's comment. To be specific, the labels have been corrected.

**((Minor comment #5))**

Line 929: Specify how to interpret the scores in the figure caption (i.e., the higher the score, the better). The same comment applies to the other figures.

**((Reply))**

The interpretation for the scores is suggested in Figures 4, 5 and 8.

**((Minor comment #6))**

Line 990: Improve the readability of the figure (there is some text outside the circles).

**((Reply))**

The text has been modified based on the reviewer's recommendation (see Line 1014).